# RoPECraft: Training-Free Motion Transfer with Trajectory-Guided RoPE Optimization on Diffusion Transformers

*Ahmet Berke Gökmen[1,2]    *Yiğit Ekin[1]    *Bahri Batuhan Bilecen[1,3,4]    Aysegul Dundar[1]

[1]Bilkent University [2]INSAIT, Sofia University "St. Kliment Ohridski" [3]ETH Zurich [4]Max Planck Institute

Reference               RoPECraft

A robotic courier zips through a maze of an eco-friendly, automated warehouse, packages whizzing past on conveyor belts.

The silhouette of a ballerina leaps across a sunlit studio, delicate shadow dancing on polished hardwood floors.

A young woman riding a small skateboard through the misty rainforest.

A vintage steam locomotive rolls past a snowy station, its black iron body steaming in the frosty air.

A camel is seen walking in an abandoned, moonlit amphitheater.

A vintage biplane loops gracefully above an airfield.

Figure 1: Our method successfully transfers the motion from reference videos.

## Abstract

We propose RoPECraft, a training-free video motion transfer method for diffusion transformers that operates solely by modifying their rotary positional embeddings (RoPE). We first extract dense optical flow from a reference video, and utilize the resulting motion offsets to warp the complex-exponential tensors of RoPE, effectively encoding motion into the generation process. These embeddings are then further optimized during denoising time steps via trajectory alignment between the predicted and target velocities using a flow-matching objective. To keep the output faithful to the text prompt and prevent duplicate generations, we incorporate a regularization term based on the phase components of the reference video's Fourier transform, projecting the phase angles onto a smooth manifold to suppress high-frequency artifacts. Experiments on benchmarks reveal that RoPECraft outperforms all recently published methods, both qualitatively and quantitatively.

---

*Equal contribution. Correspondence: berke.gokmen@insait.ai. Project page

39th Conference on Neural Information Processing Systems (NeurIPS 2025).

# 1 Introduction

Diffusion transformers (DiT) have become a leading approach for conditional video generation, producing realistic and coherent content across diverse scenarios [6, 16, 20, 50, 21, 38, 44]. While text conditioning provides a convenient interface, they are often too ambiguous to specify detailed spatio-temporal dynamics such as body movement, camera motion, or interactions. As generative quality improves, so does the demand for more precise and controllable motion synthesis.

To address the limitations of text-based motion control, earlier methods introduced explicit structural cues such as masks, bounding boxes, or depth maps to guide motion [9, 43, 40]. These approaches assume consistent geometry between the reference and generated videos, which often fails under domain shifts [45]. More recent work has shifted toward leveraging latent representations within generative models. Some methods extract motion features from internal activations [14, 45, 42], while others modify the latent prior [4] to better align reference and generated motions. A prominent example, Go with the Flow (GWTF) [4], uses a pretrained optical flow model [37] to generate motion priors that warp the initial noise input while maintaining its Gaussianity. This reportedly stabilizes and speeds up convergence. However, directly warping noise disrupts the intended latent distribution of the pre-trained DiT, as seen in Fig. 2. Even with inference-time latent optimization, the method struggles with generalization and domain shifts. As a result, GWTF requires costly fine-tuning, demanding around 40 GPU days [4]. DiTFlow [31] offers a more efficient alternative by optimizing latents or positional embeddings at test time without model retraining. However, it incurs high computational costs due to its reliance on a full-size attention-based feature computation.

We extend DiTFlow's approach by updating only positional embeddings, avoiding latent space deviation and content leakage. Our method introduces motion-augmented rotary positional embeddings, warped via optical flow-derived displacements to embed motion cues (Section 4.1). We enhance this with flow-matching-guided optimization during early time steps, enabling stable and precise generation (Section 4.2). We further ensure spatiotemporal consistency through a Fourier phase regularization (Section 4.3). Unlike GWTF, our method requires no backbone training, drastically cutting computational costs. Compared to DiTFlow, it delivers higher efficiency and better motion quality. In addition, to evaluate motion alignment, we propose Fréchet Trajectory Distance (FTD) (Section 5.2). Our method outperforms recent approaches in qualitative and quantitative assessments.

Our contributions are:

- An efficient motion transfer, RoPECraft, that leverages motion-augmented rotary positional embeddings in a training-free setting, without requiring any backbone fine-tuning.

- A novel use of optical flow displacements to warp rotary positional embeddings, encoding spatial motion cues in attention calculations.

- A unified optimization strategy combining flow matching velocity prediction and phase constraint regularization to enhance motion accuracy and ensure temporal coherence.

- A new evaluation metric, Fréchet Trajectory Distance (FTD), for quantifying motion alignment between generated and reference videos.

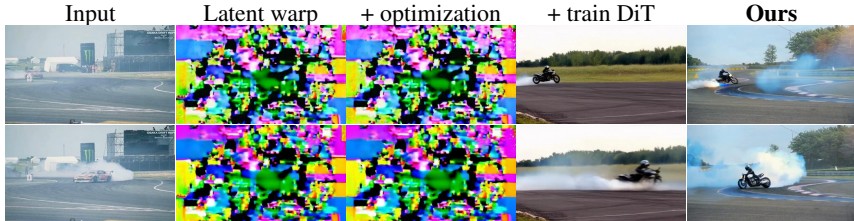

| Input | Latent warp | + optimization | + train DiT | **Ours** |

A motorcycle is seen riding on a track, kicking up smoke as it goes.

Figure 2: Latent warping [4] without an expensive fine-tuning of the DiT fails (Column 2), and latent optimization is not adequate to recover the domain shift (Column 3). Our approach keeps the latent space intact, and performs successful motion transfer, all without model re-training (Column 4).

## 2 Related Work

**Text-to-video models.** Following the successful application of diffusion models in image generation [12, 30, 29, 34], efforts have been made to extend this approach to video generation. Initial efforts used U-Net-based architectures for this purpose, utilizing temporal modules [16] or inflated convolutions [1, 3, 20, 39] to transfer the image prior to the video domain. More recently, DiT pipelines [21, 44, 38, 25, 50, 28] have gained attention due to their superior capabilities in temporal modeling and enhanced quality. Motivated by these, we also adopt a DiT backbone.

**Motion transfer.** The goal of motion transfer is to synthesize videos whose dynamics match a reference clip while disentangling motion from appearance. Earlier methods injected explicit structure (masks or depth maps) [9, 43, 40]. Subsequent work learned dedicated motion embeddings and fed them to the generator [23, 22]. Recent approaches exploit the dense motion signals already present in backbone features [42, 15, 45, 14], or condition on trajectories extracted from the reference [48, 46, 41]. The current state of the art include Go With The Flow [4], which warps the initial noise with reference flow and fine-tunes the DiT on this prior, and DiTFlow [31], which derives displacement maps from cross-frame attention and updates either latents or positional embeddings. Building on DiTFlow's insight, we dynamically update RoPE to guide attention toward reference motion while keeping the backbone frozen. Unlike prior work, we initialize RoPE with our motion-augmentation algorithm and regularizers, enabling fast and accurate motion transfer, without requiring model fine-tuning [4], inversion [42, 45, 14], masks [14], and high GPU memory during tuning [31].

**Positional embeddings in vision transformers.** Transformers lack inherent order awareness, so Vision Transformers (ViTs) [11] rely on positional embeddings to encode spatial relationships among patches. In early ViT's fixed sinusoidal or learnable absolute embeddings were used [11, 8]. These failed to generalize across varying input resolutions or sequence lengths in videos [8, 13]. Rotary Position Embedding (RoPE) overcomes these issues by rotating query and key values according to patch positions, thereby capturing relative spatial or temporal relationships [35] . Originally successful in language models [35, 2], RoPE has since been adapted to vision models [27, 17, 29, 21, 44, 38]. Building on these advances, our method updates RoPE embeddings on the fly during generation.

## 3 Preliminaries

### 3.1 Flow matching

Flow Matching (FM) is a generative modeling approach that learns a deterministic, time-dependent velocity field to transform a simple base distribution into a complex target distribution [26]. Unlike diffusion models, which reverse stochastic processes [19], FM minimizes the discrepancy between the model velocity $v_\theta(t, x)$ and a target velocity $u_t(x)$ derived from the continuity equation:

$$\mathcal{L}_{\text{FM}} = \mathbb{E}_{t, x \sim p_t} \| v_\theta(t, x) - u_t(x) \|^2 \tag{1}$$

This ensures mass-preserving transport along a predefined probability path $p_t$, enabling more efficient training and sampling than traditional diffusion models.

### 3.2 Rotary position embeddings (RoPE)

RoPE [35] encodes position by rotating query and key values $\mathbf{x}$ in the complex plane, enabling the model to capture relative positional relationships. Given a token at position $m$ with vector $\mathbf{x}_m \in \mathbb{R}^d$, the vector is split into $d/2$ pairs. Each pair $(\mathbf{x}_m^{(2i-1)}, \mathbf{x}_m^{(2i)})$ is interpreted as a complex number $\mathbf{z}_m^{(i)} = \mathbf{x}_m^{(2i-1)} + j\,\mathbf{x}_m^{(2i)}$, and RoPE applies the rotation $\mathbf{z}_m^{(i)} \cdot \Phi_{m,i}$, where $\Phi_{m,i} = e^{jm\,\theta^{-2i/d}}$ is constructed by Algorithm 1, and $\theta$ is the base frequency. This operation embeds position into the phase of each frequency component, enabling self-attention to capture relative positions through inner products of queries and keys. Since attention patterns can control motion, we leverage RoPE heavily for our motion transfer task.

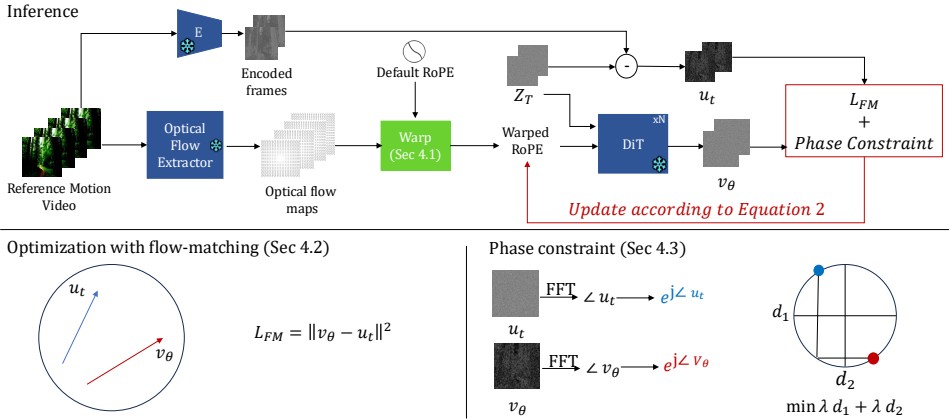

Figure 3: Visual description of our proposed pipeline inference and RoPE optimization approach.

# 4 Methodology

This section thoroughly explains the proposed components of our motion transfer method. The overall architecture is given in Fig. 3. We augment the RoPE tensors via optical flow maps (Section 4.1), and optimize them during the generation process (Section 4.2) with additional constraints (Section 4.3).

## 4.1 Motion-augmented RoPE

---

**Algorithm 1** Default 1D RoPE, expanded to 3D

---

1: **Input:** Base frequency $\theta \in \mathbb{R}_{>0}$
2: Embedding dims $D_t, D_h, D_w \in \mathbb{N}$
3: Sequence lengths $S_t, S_h, S_w \in \mathbb{N}$
4: **for** each $k \in \{t, h, w\}$ **do**
5: $\quad \mathbf{p} = [0, 1, \ldots, S_k - 1]^{\mathrm{T}}$ $\qquad \rhd \in \mathbb{R}^{S_k}$
6: $\quad \mathbf{d} = [0, 1, \ldots, D_k/2 - 1]^{\mathrm{T}}$ $\quad \rhd \in \mathbb{R}^{D_k/2}$
7: $\quad \mathbf{f} = \theta^{-2\mathbf{d}/D_k}$ $\qquad\qquad \rhd \in \mathbb{R}^{D_k/2}$
8: $\quad \mathbf{\Phi}_k = e^{j\mathbf{p}\mathbf{f}^{\mathrm{T}}}$ $\qquad\qquad \rhd \in \mathbb{C}^{S_k \times (D_k/2)}$
9: $\quad \mathbf{\Phi}_k = \texttt{expand}(\mathbf{\Phi}_k)$ $\rhd \in \mathbb{C}^{S_t \times S_h \times S_w \times (D_k/2)}$
10: **end for**
11: $\mathbf{\Phi} = \texttt{concat}(\mathbf{\Phi}_{t,h,w})$ $\qquad \rhd \in \mathbb{C}^{S_t \times S_h \times S_w \times (D/2)}$
12: $\mathbf{\Phi} = \texttt{flatten}(\mathbf{\Phi})$ $\qquad \rhd \in \mathbb{C}^{1 \times 1 \times (S_t S_h S_w) \times (D/2)}$

---

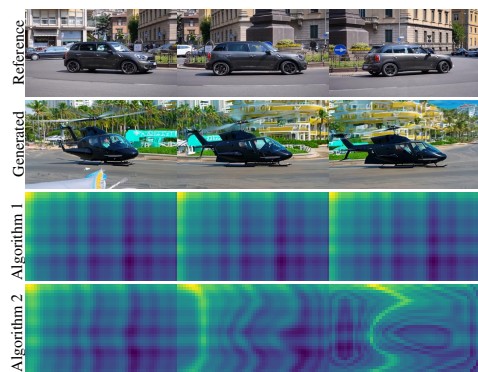

Figure 4: Comparison of the generations of default and motion-augmented RoPE.

The default RoPE algorithm used in video-DiTs is presented in Algorithm 1, where standard 1D RoPE is independently applied along the temporal ($t$), height ($h$), and width ($w$) dimensions to produce the respective components $\mathbf{\Phi}_t$, $\mathbf{\Phi}_h$, and $\mathbf{\Phi}_w$. These are then combined to form the full 3D positional encodings $\mathbf{\Phi}$, where each dimension $k \in \{t, h, w\}$ in $\mathbf{\Phi}_k$ is expanded (repeated) in the other two dimensions, $\{t, h, w\} \setminus k$. However, as previously discussed, our insight is that this formulation can be altered significantly with motion signals. Specifically, by having unique, motion-augmented 1D RoPEs for $h$ and $w$ components, we allow the attention mechanism during the generation process to better understand which spatial patches should attend to one another.

Our proposed procedure is detailed in Algorithm 2. For each row and column, we use the processed motion signals $\mathbf{h}_{\text{flow}}$ and $\mathbf{w}_{\text{flow}}$ to adjust the positional indices $\mathbf{p}$ in the complex exponential $\mathbf{\Phi} = \exp(j\mathbf{p}\mathbf{f}^{\mathrm{T}})$. Unlike Algorithm 1, where $\mathbf{\Phi}_h$ is fixed across all rows, and $\mathbf{\Phi}_w$ across all columns, Algorithm 2 introduces variation based on the motion signals. This way, we can construct unique embeddings for each spatial row and column for $\mathbf{\Phi}_h$ and $\mathbf{\Phi}_w$, respectively, providing a better initial condition for a motion-guided generation. We leave the temporal component $\mathbf{\Phi}_t$ unmodified, as altering it often introduces decoding artifacts without significant benefit.

Fig. 4 compares the default (Row 3) and modified (Row 4) embeddings visually. It can be observed that Algorithm 2 warps RoPE tensors in the motion direction (Row 1), whose effects are reflected on the generation (Row 2). Fig. 5 demonstrates the effectiveness of our approach across various prompts, which transfers coarse input motion directly into the generated videos.

| Input | Output | Input | Output | Input | Output |
|---|---|---|---|---|---|

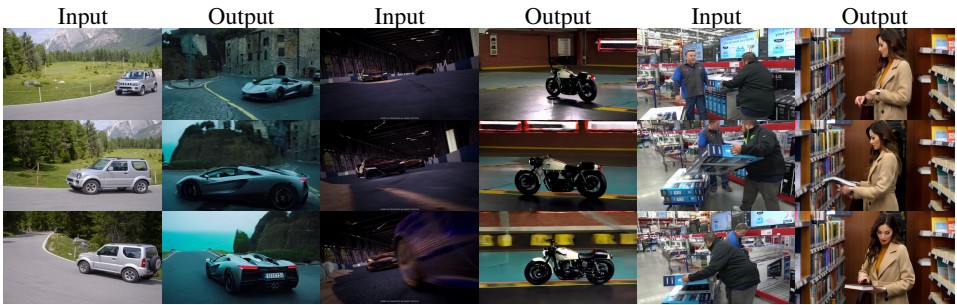

Figure 5: Qualitative results of motion-augmented RoPE described in Algorithm 2.

However, relying solely on the modification introduced in Algorithm 2 can yield suboptimal results. For example, while the overall motion may be correct, subjects sometimes face the opposite direction (Column 4) or fail to accurately follow challanging trajectories (Column 6). To address these limitations, we introduce a brief optimization step over the motion-augmented RoPE tensors during generation, which will be described in Section 4.2.

## 4.2 Optimization with flow-matching

---

**Algorithm 2** Motion-augmented RoPE

---

1: **Input:** Base frequency $\theta \in \mathbb{R}_{>0}$,
2: Embedding dims $D_t, D_h, D_w \in \mathbb{N}$
3: Sequence lengths $S_t, S_h, S_w \in \mathbb{N}$
4: Optical flows $\mathbf{u}, \mathbf{v}$ $\qquad \triangleright \in \mathbb{R}^{2 \times S_t \times H \times W}$

5: $\mathbf{u}, \mathbf{v} = \texttt{downsample}(\mathbf{u}, \mathbf{v})$ $\quad \triangleright \in \mathbb{R}^{2 \times S_t \times S_h \times S_w}$
6: $\mathbf{h}_{\text{flow}}, \mathbf{w}_{\text{flow}} = \texttt{cumsum}(\mathbf{u}, \mathbf{v})$
7: $\mathbf{h}_{\text{flow}} = \texttt{flatten}(\mathbf{h}_{\text{flow}})$ $\quad \triangleright \in \mathbb{R}^{(S_t \times S_w) \times S_h}$
8: $\mathbf{w}_{\text{flow}} = \texttt{flatten}(\mathbf{w}_{\text{flow}})$ $\quad \triangleright \in \mathbb{R}^{(S_t \times S_h) \times S_w}$

9: $\mathbf{f}_h = \theta^{-2[0,1,\dots,D_h/2-1]^{\text{T}}/D_h}$ $\qquad \triangleright \in \mathbb{R}^{D_h/2}$
10: **for** each $r$ in $[0, 1, \dots, S_t \times S_w]$ **do**
11: $\quad \mathbf{p} = [0, 1, \dots, S_h - 1]^{\text{T}} + \mathbf{h}_{\text{flow}}[r]$ $\triangleright \in \mathbb{R}^{S_h}$
12: $\quad \boldsymbol{\Phi}_h[r] = e^{j\mathbf{p}\mathbf{f}_h^{\text{T}}}$ $\qquad \triangleright \in \mathbb{C}^{S_h \times (D_h/2)}$
13: **end for**
14: $\boldsymbol{\Phi}_h = \texttt{reorder}(\boldsymbol{\Phi}_h)$ $\quad \triangleright \in \mathbb{C}^{S_t \times S_h \times S_w \times (D_h/2)}$

15: $\mathbf{f}_w = \theta^{-2[0,1,\dots,D_w/2-1]^{\text{T}}/D_w}$
16: **for** each $c$ in $[0, 1, \dots, S_t \times S_h]$ **do**
17: $\quad \mathbf{p} = [0, 1, \dots, S_w - 1]^{\text{T}} + \mathbf{w}_{\text{flow}}[c]$
18: $\quad \boldsymbol{\Phi}_w[c] = e^{j\mathbf{p}\mathbf{f}_w^{\text{T}}}$
19: **end for**
20: $\boldsymbol{\Phi}_w = \texttt{reorder}(\boldsymbol{\Phi}_w)$

21: $\mathbf{p} = [0, \dots, S_t - 1]^{\text{T}}$
22: $\mathbf{f} = \theta^{-2[0,\dots,D_t/2-1]^{\text{T}}/D_t}$
23: $\boldsymbol{\Phi}_t = \texttt{expand}(e^{j\mathbf{p}\mathbf{f}^{\text{T}}})$

24: $\boldsymbol{\Phi} = \texttt{flatten}(\texttt{concat}(\boldsymbol{\Phi}_{t,h,w}))$

---

To refine Algorithm 2, we apply a brief optimization on rotary embeddings during early generation steps. Using Eq. (1), we align the generated velocity $v_\theta(t, x_t)$ with the target velocity $u_t(x) = \sigma_t^{-1}(x_t - \mathbf{v})$, where $x_t$ is the current latent in time step $t$, $\mathbf{v}$ is latent reference video, and $\sigma$ is the scheduler sigma.

Fig. 6 illustrates the effectiveness of both optimization and the motion-augmented RoPE initial condition. In Column 1, the subject moves away from the camera, while in Column 2, the subject moves from left to right. The motion-augmented RoPE approach (Columns 3–4) successfully captures the general movements. However, in the second sample, it incorrectly renders the motorbike facing backward. When optimization is performed without a dedicated initial condition (Algorithm 1), the subject placement improves, but issues arise in motion direction (Column 6), and visual artifacts appear (Column 5, Row 2). In contrast, initializing with Algorithm 2 yields the best results across the samples. This approach reduces artifacts, corrects subject orientation and trajectories.

## 4.3 Phase constraints

Flow matching optimization produces strong results, as shown in Fig. 6, but we occasionally observe duplicated subjects when adjusting the orientation, position, or motion of the moving subjects. To address this, we build on insights from prior work [47] and analyze the Fourier transform of our signals. Since linear displacements in the spatial domain cause a phase shift in frequency domain,

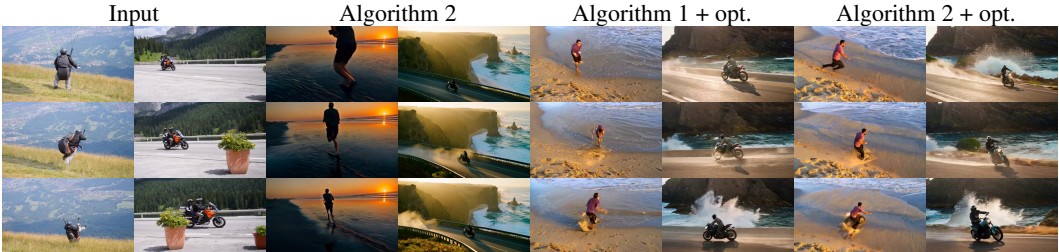

| Input | Algorithm 2 | Algorithm 1 + opt. | Algorithm 2 + opt. |

Figure 6: Qualitative results on optimization, with identical seeds across different experiments.

a Fourier property closely tied to motion transfer, we add a phase constraint to the flow matching objective to guide the model toward more accurate and consistent spatiotemporal alignment.

Specifically, we take the Fourier transform of the target velocity $u_t$ along spatio-temporal dimension, to get $\mathcal{F}(u_t) = \mathbf{U_t} = |\mathbf{U_t}| \exp\{j\angle\mathbf{U_t}\}$, where $|\mathbf{U_t}| = \{\mathfrak{Re}(\mathbf{U_t})^2 + \mathfrak{Im}(\mathbf{U_t})^2\}^{1/2}$ is the magnitude, and $\angle\mathbf{U_t} = \arctan\{\mathfrak{Im}(\mathbf{U_t})/\mathfrak{Re}(\mathbf{U_t})\}$ is the phase. We perform the same transform to the DiT output $v_\theta$, and add the phase constraint as a $\mathcal{L}_1$ regularizer to the main optimization objective. We represent the phase on the unit circle ($\exp\{j\angle\mathcal{F}(\cdot)\}$) to make them continuous and differentiable everywhere, since the original mapping $\angle\mathcal{F}(\cdot)$ contains jump discontinuities at $\pm\pi$ due to being bounded by $(-\pi, \pi]$. More clearly, we represent $\exp\{j\angle\mathcal{F}(\cdot)\} = \cos(\angle\mathcal{F}(\cdot)) + j\sin(\angle\mathcal{F}(\cdot))$ and perform the phase-consistency loss in two parts. The final optimization objective is given in Eq. (2):

$$\min \mathcal{L}_{\text{FM}}(u_t, v_\theta) + \lambda\|\cos\angle\mathcal{F}(u_t) - \cos\angle\mathcal{F}(v_\theta)\| + \lambda\|\sin\angle\mathcal{F}(u_t) - \sin\angle\mathcal{F}(v_\theta)\|, \quad (2)$$

where $\lambda$ is the hyperparameter.

Fig. 7 reveals the effect of phase constraints, fixing duplicate generations and artifacts. Additional experiments regarding Fourier components are given in Supplementary.

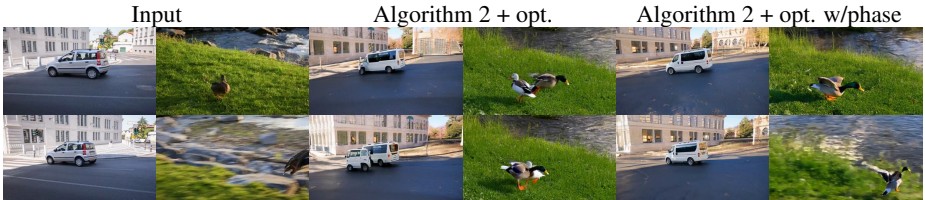

| Input | Algorithm 2 + opt. | Algorithm 2 + opt. w/phase |

Figure 7: Qualitative results on phase constraints, with identical seeds across different experiments.

## 5 Experimental Results

### 5.1 Metrics and baselines

We compare our method with the recently published motion transfer methods [4, 14, 31, 45, 42]. For evaluation, we use videos from the DAVIS dataset [32]. We generate 4 diverse prompts per DAVIS video via ShareGPT4V [7] and LLAMA 3.2 [36], which are detailed in the Supplementary.

For the evaluation, we use content-debiased Fréchet Video Distance (CD-FVD) [5] for evaluating fidelity, CLIP similarity [18] for evaluating frame-wise prompt fidelity using ViT B/32 model [33], and Motion Fidelity (MF) [45] along with our proposed metric Fréchet Trajectory Distance (FTD) for evaluating the motion alignment between the generated and the ground truth reference motion video. For assessing the motion of the foreground object as well as camera motion, we use FTD by only sampling from the foreground object mask region, and sampling from both the foreground and background mask region. For MF and FTD, the trajectories are obtained using Co-Tracker3 [24].

For video synthesis, we use Wan2.1-1.3B [38] as the backbone of our method. To obtain a fair assessment, similar to the approach used in DiTFlow [31], we adapt MOFT [42], SMM [45], DitFlow [31] and ConMo [14] to Wan2.1. For the methods that require DDIM inversion [42, 45, 14],

we perform KV-injection from reference video latents similar to [31]. We evaluate GWTF using their CogVideoX-2B [21] checkpoint. The hyper parameters are detailed in Supplementary.

## 5.2 Fréchet Trajectory Distance

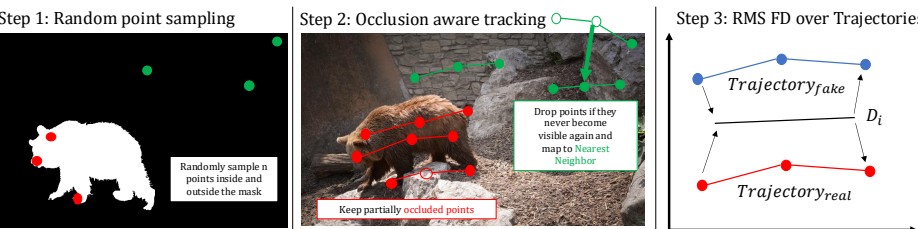

Figure 8: **Fréchet Trajectory Distance (FTD). 1)** Sample $n$ foreground (**red**) and $n$ background (**green**) seeds on the first frame. **2)** Track each seed with an occlusion-aware filler: copy the nearest visible neighbor while occluded and discard tracks that never re-appear. **3)** Measure the RMS Fréchet distance between generated (**fake**) and reference (**real**) tracks.

**Discrete Fréchet Distance.** Let $\mathbf{x}_{i,t} \in \mathbb{R}^2$ be the 2D image coordinate of the $i^{\text{th}}$ point at frame $t$ ($1 \leq t \leq T$). We denote the reference and generated trajectories by $\mathcal{T}_i^{\text{real}} = \{\mathbf{x}_{i,1}^{\text{real}}, \dots, \mathbf{x}_{i,T}^{\text{real}}\}$ and $\mathcal{T}_i^{\text{fake}} = \{\mathbf{x}_{i,1}^{\text{fake}}, \dots, \mathbf{x}_{i,T}^{\text{fake}}\}$, respectively. Then, the discrete Fréchet distance is defined as:

$$D_F\left(\mathcal{T}_i^{\text{real}}, \mathcal{T}_i^{\text{fake}}\right) = \min_{\sigma,\tau:\{1,\dots,L\}\mapsto\{1,\dots,T\}} \max_{k=1,\dots,L} \left\|\mathbf{x}_{i,\sigma(k)}^{\text{real}} - \mathbf{x}_{i,\tau(k)}^{\text{fake}}\right\|_2, \tag{3}$$

where $L$ is the length of the common re-parameterization, and $(\sigma, \tau)$ are non-decreasing index maps that allow each curve to pause or advance but never step backwards. The inner $\max$ takes the worst spatial gap along a particular pairing of frames, while the outer $\min$ selects the pairing that makes this worst gap as small as possible. Consequently, $D_F$ is the minimal worst-case deviation between the trajectories after they are aligned in time as favorably as the monotone constraint permits.

**Fréchet Trajectory Distance (FTD).** We utilize Eq. (3) in our proposed FTD metric, along with several tricks to obtain meaningful trajectories $\mathcal{T}$. Fig. 8 explains our procedure thoroughly. From the first frame, we uniformly select $n$ points inside the binary foreground mask $\mathcal{M}_0$, and $n$ outside to capture both object (red) and background (green) motion. Then, the occlusion-aware tracker [24] generates tracks starting with the initial $2n$ points. Since some points may go out of bounds or get occluded as the video progresses, we reassign them by copying to their nearest visible neighbor to maintain trajectory continuity, and drop a track entirely if the associated point never reappears. Before computing distances, all coordinates are normalized by the frame width $W$ and height $H$, making the metric resolution-invariant. The procedure yields $N$ valid, temporally coherent pairs $\{\mathcal{T}_i^{\text{real}}, \mathcal{T}_i^{\text{fake}}\}_{i=1}^N$. Utilizing the pairs, we calculate root-mean-square Fréchet distance, FTD $= \left(N^{-1}\Sigma_{i=1}^N D_F^2(\mathcal{T}_i^{\text{real}}, \mathcal{T}_i^{\text{fake}})\right)^{0.5}$. For calculating $D_F$, we utilize [10].

**Comparison with Motion Fidelity [45].** The Motion Fidelity (MF) metric [45] computes cosine similarity between frame-to-frame displacements on a fixed grid, averaging best matches. However, it ignores path shape, magnitude, and occlusions, and can report high scores even when trajectories diverge. In contrast, our Fréchet Trajectory Distance (FTD) drops unreliable tracks, focuses on relevant regions, and measures curve distance using discrete Fréchet distance, making it more robust to missing data and outliers. As shown in Table 1, MF also exhibits much higher standard deviation, highlighting its instability compared to FTD.

To further illustrate the behavioral difference between the two metrics, we present two controlled toy examples in Fig. 9. In Case 1, the generated trajectory follows the same path as the reference but with a smaller motion magnitude, while in Case 2, both trajectories share identical motion directions yet are spatially offset. Because the Motion Fidelity (MF) metric normalizes motion vectors to unit length and computes directional cosine similarity over a fixed grid, it reports perfect similarity (MF=1.0) in both cases. This normalization causes MF to ignore motion speed, path magnitude, and spatial alignment, effectively making it translation-invariant but insensitive to geometric deviation. Moreover, MF can be artificially inflated by nearest-neighbor matches anywhere in the frame, as it measures

average directional alignment between motion vectors rather than actual trajectory correspondence. In contrast, the Fréchet Trajectory Distance (FTD) evaluates the geometric trajectory consistency over time by measuring the curve distance between corresponding paths. FTD therefore penalizes drift, speed changes, and path-shape differences directly. FTD increases proportionally with geometric and temporal discrepancies (FTD=1.414 and 1.0 for Cases 1 and 2, respectively), demonstrating its discriminative power and perceptual robustness.

## 5.3 Qualitative evaluation

Case 1: Same trajectory & Different Magnitude  Case 2: Same trajectory & Different Offset

FTD: 1.414 | MF: 1.0      FTD: 1.0 | MF: 1.0

●— Generated      ●— Original

Figure 9: Illustration of the behavioral difference between Motion Fidelity (MF) and Fréchet Trajectory Distance (FTD) across two controlled toy cases. In both cases, the generated trajectories (red) differ from the original trajectories (blue) either in magnitude (Case 1) or in spatial offset (Case 2). Although both pairs exhibit substantial geometric deviation, MF remains artificially high because it measures only the directional alignment of motion vectors, discarding scale and positional information. In contrast, FTD penalizes such discrepancies by accounting for the full geometric path similarity, thereby providing a more faithful measure of motion consistency. It is important to note that FTD evaluates distance, hence smaller values mean better motion alignment where MF evaluates direct motion alignment, hence larger values is better.

Fig. 10 provides a visual comparison of the evaluated methods across diverse prompts and motion scenarios. Our approach consistently outperforms others in both trajectory direction and subject orientation. In P1, MOFT, DitFlow, ConMo, and SMM fail to capture the correct motion direction, although SMM maintains proper subject orientation. In P2, some methods struggle with prompt alignment, such as keeping the man stationary, and GWTF introduces noticeable artifacts. For more complex motions like P3 and P4, most methods do not use the reference motion effectively. While GWTF shows motion coherence, it often sacrifices from prompt alignment. For example, it merges a motorcycle with a truck in P3 and does not place the man walking on the wooden dock in P2. A similar issue appears in P6, where GWTF generates a distorted motorhome, and only SMM, GWTF, and our method reflect the reference motion correctly. In general, our method accurately captures both motion and subject across all examples. Additional results of our model on challenging videos such as videos with camera motion or multiple subjects can be seen in the Section A.6.

## 5.4 Quantitative evaluation

Table 1 compares our method with recent motion transfer baselines across five key metrics: Motion Fidelity (MF) [45], content-debiased Fréchet Video Distance (CD-FVD) [5], CLIP similarity [18], and our occlusion-aware FTD on foreground (FG) and all points (FG+BG).

Our method achieves the highest MF score (**0.5816**) and the lowest CD-FVD (**1284.58**), surpassing a strong baseline, GWTF [4], by +0.0103 (approximately +1.8%) and -200.6 (approximately -13.5%), respectively. It also attains the second-best CLIP similarity (**0.2350**), and ranks second on both FTD variants, **0.2644** for FG and **0.2584** for FG+BG, while outperforming all remaining competitors. In terms of runtime, our method runs at **109.231 ± 3.112 s**, comparable to SMM [45] (107.281 ± 4.562 s), DitFlow (RoPE) [31] (104.405 ± 2.056 s), DitFlow (Latent) [31] (105.126 ± 2.739 s), MOFT [42] (119.411 ± 3.847 s), and GWTF [4] (101.342 ± 3.337 s), while being significantly faster than ConMo [14] (150.666 ± 3.317 s). This demonstrates that our framework achieves a strong trade-off between computational efficiency and high-fidelity generation quality.

We also showcase quantitative scores in ablation studies, validating the effectiveness of our design. Specifically, Table 2 justifies our approach on motion-augmented RoPE, optimization procedure with flow-matching, and phase constraints. Table 3 elaborates on the selection of first $t$ denoising steps in our optimization, and $s$ number of optimization steps per $t$. We opted for $(t, s) = (10, 5)$ as we noticed that $s = 10$ decreases visual quality significantly.

Reference    **Ours**    GWTF    SMM    MOFT    DitFlow    ConMo

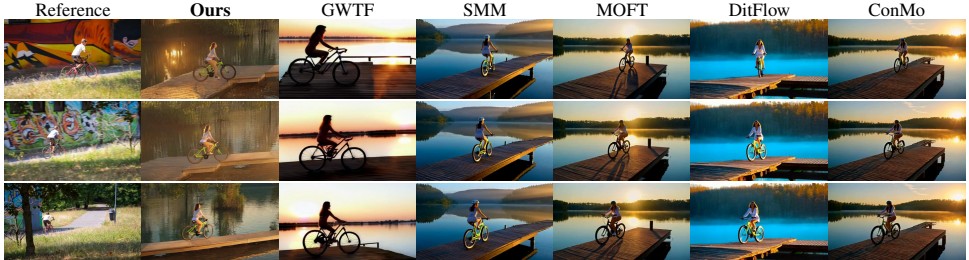

P1: A woman rides a bike down a wooden dock alongside a serene lake at sunrise.

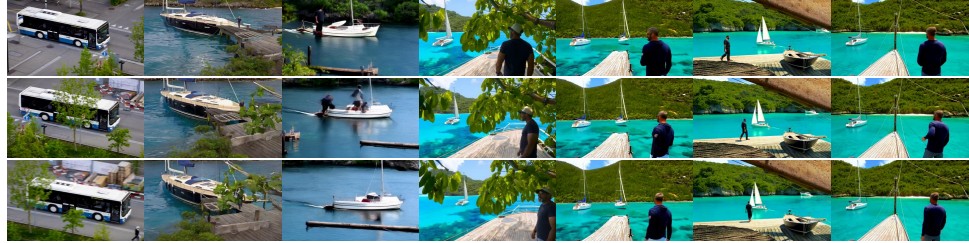

P2: A sailboat is anchored in a tranquil cove surrounded by greenery, as the man walks along the weathered wooden dock.

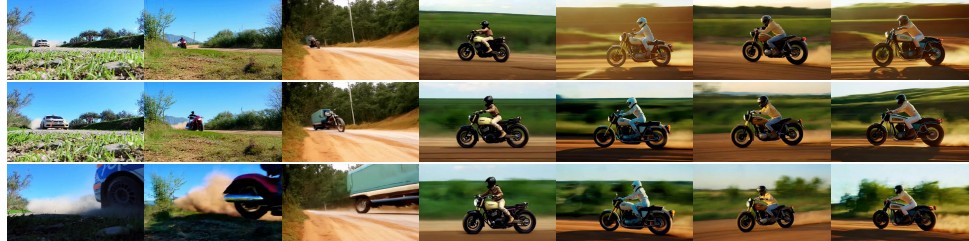

P3: The motorcycle drives down a dirt road, and then speeds up as it goes.

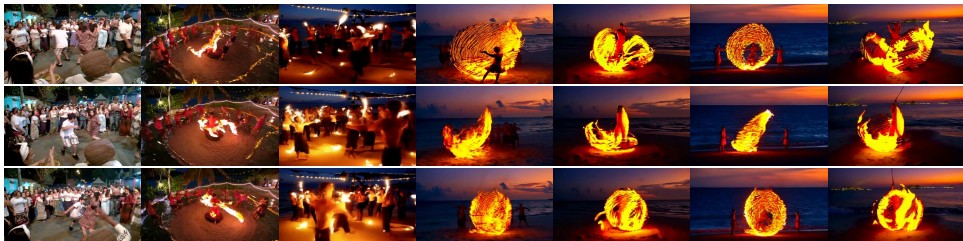

P4: A large group of fire dancers are spinning together in a circle, with one performer leading in middle, on a tropical beach.

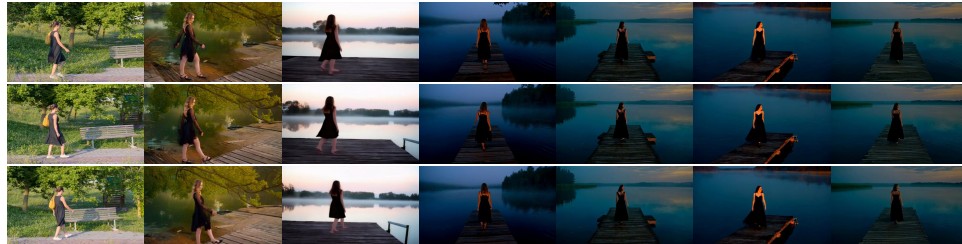

P5: A woman wearing a black dress walks down a worn wooden dock along the edge of a misty lake at dusk.

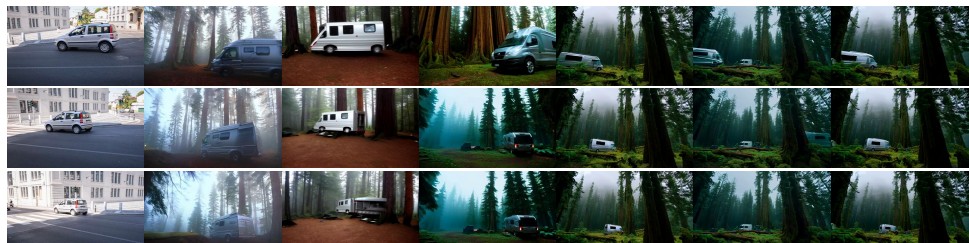

P6: A silver motorhome pulling up to a campsite nestled among the towering redwoods of a misty forest.

Figure 10: Qualitative comparison of the methods with diverse prompts.

Table 1: Comparison of motion transfer methods across evaluation metrics. Best and second results are represented with *italic* and underlined, respectively.

| Method | MF ↑ | CD-FVD ↓ | CLIP ↑ | FTD (FG) ↓ | FTD (FG+BG) ↓ |
|---|---|---|---|---|---|
| GWTF [4] | 0.5713±0.22 | 1485.23 | *0.2378±0.04* | *0.2457±0.14* | *0.2308±0.10* |
| SMM [45] | 0.4889±0.20 | 1600.33 | 0.2331±0.04 | 0.2882±0.15 | 0.3176±0.15 |
| MOFT [42] | 0.4606±0.20 | 1630.45 | 0.2311±0.04 | 0.2811±0.16 | 0.3057±0.14 |
| DitFlow (latents) [31] | 0.4832±0.20 | 1735.49 | 0.2339±0.04 | 0.2921±0.15 | 0.3135±0.12 |
| DitFlow (RoPE) [31] | 0.4500±0.18 | 1852.90 | 0.2345±0.04 | 0.2785±0.14 | 0.3019±0.13 |
| ConMo [14] | 0.4627±0.21 | 1680.78 | 0.2309±0.04 | 0.2769±0.15 | 0.3040±0.14 |
| **Ours** | *0.5816±0.19* | *1284.58* | 0.2350±0.04 | 0.2644±0.14 | 0.2584±0.13 |

Table 2: Ablation on motion-augmented RoPE and phase constraints.

| Method | MF | CLIP | FTD |
|---|---|---|---|
| Alg.1 + opt. | 0.7082 | 0.1560 | 0.2174 |
| Alg.2 + opt. | 0.7092 | 0.1650 | 0.2105 |
| Alg.2 + opt. + phase | *0.7210* | *0.1656* | *0.2060* |

Table 3: Ablation on hyperparameters.

| $(t, s)$ | MF | CD-FVD | CLIP | FTD |
|---|---|---|---|---|
| 5, 5 | 0.5165 | 1437.99 | 0.1597 | 0.2901 |
| 5, 10 | 0.5523 | 1606.88 | 0.1663 | 0.2728 |
| 10, 5 | 0.5675 | *1364.25* | *0.1664* | 0.2633 |
| 10, 10 | *0.6160* | 1492.86 | 0.1572 | *0.2573* |

## 5.5 User study

We conducted a user study to evaluate (i) how well our proposed FTD and MF metrics align with human perception, and (ii) overall method quality. We randomly sampled 20 prompt-reference pairs from DAVIS and generated outputs for all competing methods. In the first part of the survey, participants selected the top-3 videos that best matched the reference motion. As shown in Table 4, FTD exhibits a noticeably stronger correlation with human motion judgments than MF. In the second part, users ranked their top-3 videos based on overall visual preference. The results in Table 5 show that our method is consistently preferred across all evaluated dimensions.

Table 4: Alignment of motion preference.

| | FTD (%) | MF (%) |
|---|---|---|
| 1st choice | **25** | 11 |
| 2nd choice | **17** | 17 |
| 3rd choice | **19** | 11 |

Table 5: User preference study for visual quality.

| Method | RPC | GWTF | SMM | MOFT | ConMo | DF-L | DF-R |
|---|---|---|---|---|---|---|---|
| 1st | **30%** | 10% | 13% | 19% | 10% | 12% | 6% |
| 2nd | **23%** | 12% | 14% | 11% | 13% | 13% | 14% |
| 3rd | 13% | 11% | 22% | 13% | 16% | 13% | 12% |

These findings reveal two key outcomes: (1) FTD aligns better with human motion perception than MF, and (2) RoPECraft is consistently preferred over all baselines in overall quality.

## 6 Conclusion and Discussion

In this paper, we introduce RoPECraft, a training-free motion transfer method that manipulates rotary positional embeddings in diffusion transformers. By combining motion-augmented RoPE tensors, flow-matching-based optimization, and phase-based regularization, RoPECraft achieves high-quality performance across multiple benchmarks, and produces high-quality motion transfer results.

For the future work, the motion-augmented RoPE framework can be extended to handle more challenging cases, such as handling motion with extreme occlusion, and better high-frequency details in the generated videos. In addition, the pipeline can be extended to controllable video editing.

We discuss limitations and broader impacts in the Supplementary Material.

**Acknowledgements.** We acknowledge EuroHPC Joint Undertaking for awarding the project ID EHPC-AI-2024A02-031 access to Leonardo at CINECA, Italy. We also acknowledge Fal.ai for granting GPU access.

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

# A Technical Appendices and Supplementary Material

## A.1 Ablation on Fourier Features

We ablate the Fourier components for additional constraints on the flow-matching objective on our optimization stage. For the objective $\min \mathcal{L}_{\text{FM}} + \mathcal{L}_{\text{c}}$, we ablate two regularizer for magnitude and phase, Eq. (4) and Eq. (5), respectively,

$$\mathcal{L}_{\text{c}} = \lambda \, \| |\mathcal{F}(u_t)| - |\mathcal{F}(v_\theta)| \|_1 \tag{4}$$

$$\mathcal{L}_{\text{c}} = \lambda \| \cos \angle \mathcal{F}(u_t) - \cos \angle \mathcal{F}(v_\theta) \| + \lambda \|_1 \sin \angle \mathcal{F}(u_t) - \sin \angle \mathcal{F}(v_\theta) \|_1, \tag{5}$$

where $u_t$ denotes the target velocity, and $v_\theta$ is the generated velocity output from the transformer at time step $t$. As shown in Fig. 11, the phase-based constraint proves more effective than the magnitude-based one, supporting the discussion in the main paper. $\lambda$ is chosen as 1.0 across all experiments, as sweeping $\lambda$ did not result in significant changes.


Reference            w/Magnitude            w/Phase


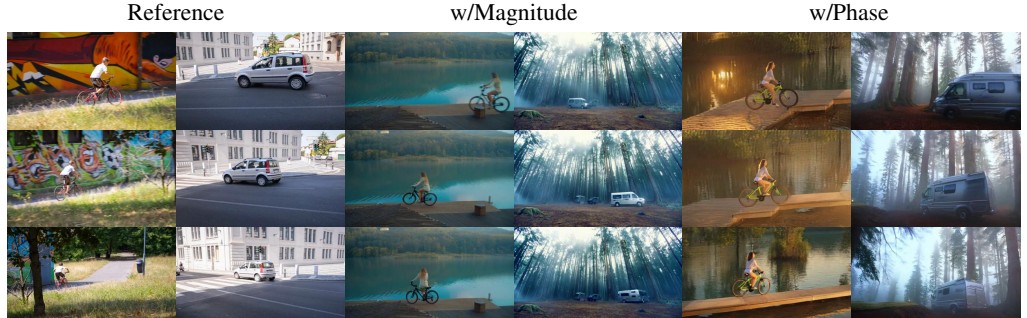

P1: A woman rides a bike down a wooden dock alongside a serene lake at sunrise.
P2: A silver motorhome pulling up to a campsite nestled among the towering redwoods of a misty forest.

Figure 11: Qualitative comparison of using magnitude and phase constraints. First column shows the video associated with P1, whereas the second column is with P2.

## A.2 Prompts

We extract prompts from DAVIS [32] videos by using ShareGPT4V [7]. For generating diverse prompts, we utilize LLAMA 3.2 [36]. We construct different `system_prompts` for changing the object (1), and environment (2). We also provide a paraphrased prompt for reconstruction (3). The system prompts are listed below:

```
system_prompt_1 = Answer with a single sentence.  You will receive a
single-sentence or multi-sentence video prompt.  Replace its main subject
(the actor or object performing the action) with a new, physically
plausible subject while leaving the action, environment, camera movements,
and style intact.  The new subject must be realistic in the described
scenario (e.g., a golden retriever a border collie; a sports car a vintage
motorcycle).  Return only the fully rewritten prompt-no explanations, no
bullet points.  Keep the scene information in the prompt the same.  Do
not just change the gender etc.  like man-woman or woman-man.  Change the
entity class, do not just replace person with person.

system_prompt_2 = Answer with a single sentence.  You will receive a
video prompt.  Keep the subject(s) and their actions exactly the same, but
relocate the scene or setting to a coherent, vivid new environment.  Ensure
lighting, weather, and background details match the new setting and remain
physically reasonable.  Output the updated prompt text and nothing else.

system_prompt_3 = Answer with a single sentence.  Do not alter the subject,
action, or scene.  Simply rephrase the text so it is stylistically
```

```
different (synonyms, varied sentence structure) while preserving every
factual detail.  Return the single paraphrased prompt-no commentary, no
headings.
```

For each original prompt from ShareGPT4V [7], we apply these system prompts to generate its corresponding response. The motion prompts utilized in the main paper figures are listed below:

- ```
  A sleek, black helicopter is seen around a bustling beach side
  promenade, passing by a seaside resort building.
  ```
- ```
  A sleek, silver sports car is navigating through the foggy streets
  of an Italian Renaissance-era town perched on the edge of a rugged
  cliff overlooking the turquoise Mediterranean Sea.
  ```
- ```
  The video shows a vintage motorcycle driving down a track in a
  garage.
  ```
- ```
  A woman wearing a beige coat is seen browsing in a bookstore,
  examining a shelf and then selecting a book from the stack.
  ```
- ```
  A man is seen sprinting across a deserted beach at sunset, his feet
  pounding against the wet sand.
  ```
- ```
  A man on a motorcycle is seen riding down a coastal highway with
  rugged cliffs and rocky outcroppings lining the edge of the ocean,
  as sunlight catches the spray of the waves and casts a misty veil
  over the scene.
  ```
- ```
  A backpacker is seen walking on a rocky terrain with mountains in
  the background.
  ```
- ```
  A white van is seen driving down a street with a building in the
  background.
  ```
- ```
  A goose walks on grass and then flies over a river.
  ```

### A.3   Hyperparameters and Computational Requirements

**Hyperparameters.** For video synthesis, we adopt Wan2.1-1.3B [38] as the backbone of our method. Since DiTFlow [31] was originally proposed on CogVideoX [21], and MOFT [42], SMM [45], and ConMo [14] were developed on UNet-based architectures, we re-implemented all these baselines using the same Wan2.1-1.3B backbone for a fairer comparison. For methods that require DDIM inversion [42, 45, 14], we applied key-value (KV) injection into all transformer blocks during the first $t$ denoising steps, following the strategy used in DiTFlow.

We conducted extensive experiments to determine optimal hyperparameters for each method in the Wan2.1 framework. The hyperparameters are defined as follows: learning rate ($l$), transformer block index for motion feature extraction ($b$), number of optimization steps ($s$), number of early denoising steps used for optimization ($t$, out of $50$ total steps), AMF attention temperature for DitFlow ($d$), and mask fusion weight for ConMo ($w$). We utilize Adam optimizer across all methods, with their default $\beta$ parameters.

1. **DiTFlow** [31]: $l = 1 \times 10^{-4}$, $b = 10$, $s = 10$, $t = 5$, $d = 2.0$
2. **MOFT** [42]: $l = 1 \times 10^{-4}$, $b = 10$, $s = 10$, $t = 5$
3. **SMM** [45]: $l = 1 \times 10^{-4}$, $b = 5$, $s = 10$, $t = 5$
4. **ConMo** [14]: $l = 1 \times 10^{-4}$, $b = 20$, $s = 5$, $t = 10$, $w = 0.5$
5. **Ours**: $l = 1 \times 10^{-4}$, $s = 5$, $t = 10$

For ConMo, we used ground-truth DAVIS masks for the reference videos. We do not extract motion cues from internal layers of the transformer, hence $b$ is not applicable for our case.

Due to limited computational resources, we used the original CogVideoX weights provided by the authors for GWTF [4], as training the full Wan2.1 pipeline from scratch was infeasible. To align more closely with Wan's 1.3B parameter scale during evaluation, we used the 2B checkpoint of GWTF.

**Computational Requirements.** We run all the models on a shared cluster, with compute nodes equipped with $4\times$ NVIDIA A100 64GB.

### A.4 Fréchet Trajectory Distance

We provide our Fréchet Trajectory Distance pseudocode in Listing 1, where the `frechetdist` is calculated by using [10] and `cotracker3` by [24].

---

**Listing 1** Fréchet Trajectory Distance implementation.

```
1  import frechetdist
2  def fill_and_drop(track, vis):
3      filled = track.clone()
4      N, F, _ = filled.shape
5      for t in range(1, F):
6          inv_idx = (~vis[:, t]).nonzero(as_tuple=False).view(-1)
7          vis_idx = vis[:, t].nonzero(as_tuple=False).view(-1)
8          if inv_idx.numel() and vis_idx.numel():
9              prev_pts = filled[inv_idx, t - 1]
10             curr_pts = filled[vis_idx, t]
11             d = distance_matrix(prev_pts, curr_pts)
12             filled[inv_idx, t] = curr_pts[d.argmin(dim=1)]
13         else:
14             filled[:, t] = filled[:, t - 1]
15     dropped = (~vis[:, 1:].any(dim=1)).nonzero(as_tuple=False).view(-1)
16     return filled, dropped
17
18 def compare_trajectory_consistency(cotracker3 , video1, video2, mask,
19                                    n_points=100, use_fg_mask_only=False):
20     _, T, C, H, W = video1.shape
21     if use_fg_mask_only:
22         queries = sample_points_inside_mask_randomly(mask, n_points)
23     else:
24         queries = sample_points_from_mask_randomly(mask, fg=n_points//2,
           ↪ bg=n_points//2)
25     tracks = []
26     drops = []
27     for vid in (video1, video2):
28         pts, vis = cotracker3(vid, queries=queries)
29         pts, drop = fill_and_drop(pts[0], vis[0])
30         pts[...,0] /= W
31         pts[...,1] /= H
32         tracks.append(pts)
33         drops.append(drop)
34     sq = []
35     for i in range(tracks[0].shape[1]):
36         if i in drops[0] or i in drops[1]:
37             continue
38         P = tracks[0]
39         Q = tracks[1]
40         fd = frechetdist(P, Q)
41         sq.append(fd*fd)
42     return sqrt(mean(sq))
```

---

### A.5 Limitations and Broader Impacts

**Limitations.** We present the limitations of our method in Fig. 13. These limitations can be mitigated by utilizing a heavier DiT-based video generation model, or a higher-quality motion extractor for motion-augmented rotary embedding generation.

A limitation of our proposed Fréchet Trajectory Distance is its reliance on CoTracker3 [24], which has difficulty handling zoom in or zoom out camera motions, especially in cases where the tracking points remain stationary. This limits the accuracy of the extracted trajectories in such scenarios.

**Broader Impacts.** The ability to generate realistic motion in videos can greatly benefit fields such as animation, virtual production, education, and accessibility. However, it also introduces risks, particularly in the creation of deepfakes and other forms of synthetic media that may be used

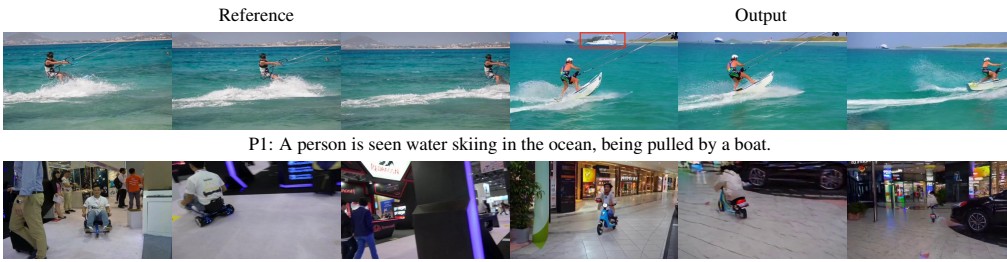

Reference                                                                    Output

P1: A person is seen water skiing in the ocean, being pulled by a boat.

P2: A figure on a scooter was observed traversing through the mall before losing balance and subsequently dropping from his vehicle.

Figure 12: Limitations of our method. In the first video, a boat (highlighted with a red rectangle) intermittently appears and disappears, likely due to limitations in the backbone network. In the second video, the final frame shows a distorted human figure, caused by the absence of the person in the corresponding frame of the source video. This may highlight a limitation of the optical flow extractor used to modify our rotary embeddings, in handling occluded or missing subjects.

to deceive. These concerns highlight the importance of responsible use, and supporting research into detection and verification methods to help mitigate potential misuse while enabling positive applications.

### A.6 More results on Challenging Videos

To further demonstrate the robustness of our method under challenging conditions, we also evaluate on a randomly sampled subsample from RealCamVid [49], each containing camera motion and multiple objects. Table 6 reveals that our method is also robust on more complex datasets than DAVIS.

Table 6: Comparison of motion transfer methods on RealCamVid [49].

| Method | MF ↑ | CLIP ↑ | FTD ↓ |
|---|---|---|---|
| GWTF | 0.5796 | 0.2087 | 0.2072 |
| ConMo | 0.5973 | 0.2014 | 0.2800 |
| SMM | 0.5717 | 0.2118 | 0.2827 |
| DiTFlow (RoPE) | 0.5686 | 0.2118 | 0.2991 |
| DiTFlow (latent) | 0.5693 | 0.2099 | 0.2962 |
| MOFT | 0.5778 | 0.2095 | 0.2873 |
| **Ours** | *0.7019* | 0.2141 | *0.1697* |

Reference             Output

P1: The video shows musicians in a studio setting with a neutral background.

P2: The video shows a pair of robots on a futuristic spaceship bridge illuminated by neon lights.

P3: The video depicts a variety of sports and luxury cars displayed inside a brightly lit showroom.

P4: The video depicts a fleet of yachts moored at a bustling marina under an orange evening sky.

P5: The video shows two cars, one purple and the other black, displayed on a rotating platform inside a showroom.

P6: The video shows two sleek yachts, one dark and the other midnight-blue.

P7: The video shows a drone race weaving through neon-lit hoops inside a dark warehouse.

P8: The video depicts a team of sled dogs pulling a musher across the snow-covered ground.

P9: The video shows a group of women in colorful dresses dancing down the street.

P10: The video captures a young boy on a bright, sunny day, walking on a sidewalk with a black metal fence, and a black cat on a leash.

P11: The video shows two individuals in a circular space with a wooden structure that has a wooden ceiling and a wooden floor.

Figure 13: Our method effectively transfers camera motion and motion from multiple subjects accurately.

