# OpenReview forum: "RoPECraft: Training-Free Motion Transfer with Trajectory-Guided RoPE Optimization on Diffusion Transformers"
_NeurIPS.cc/2025/Conference — NeurIPS 2025 poster_

### Official Review · Reviewer_d2Er · 2025-06-26

**Clarity:** 3
**Significance:** 3
**Originality:** 3
**Rating:** 4
**Confidence:** 4

**Summary:**

The authors propose a novel training-free motion transfer method. Specifically, they extract optical flows from the source video and apply a warping operation to the RoPE embedding to incorporate motion into the generation process. The warped embeddings are then optimized during the denoising steps. Additionally, they introduce a new regularization term based on the phase angle of the source video’s Fourier transform to suppress visual artifacts. Experimental results are conducted on standard benchmarks to demonstrate the effectiveness of their approach.

**Questions:**

Please refer to the weaknesses for my questions on this paper.

**Ethical Concerns:**

["NO or VERY MINOR ethics concerns only"]

**Final Justification:**

The proposed method is novel and has achieved competitive results. The authors also formulate the paper with clarity. In the rebuttal, the author has addressed my concerns regarding the runtime and hard case results. Therefore, I would like to keep my score as borderline accept.

**Limitations:**

Yes.

**Paper Formatting Concerns:**

NA.

**Quality:**

3

**Strengths And Weaknesses:**

**Strengths**
- Novelty. The authors propose a novel motion transfer method that leverages optical flow to warp rotary positional embeddings.
- Competitive results. The proposed method achieves competitive performance in both qualitative and quantitative evaluations.
- Clarity. The paper is clearly written and well-organized.

**Weaknesses**

- The paper does not include runtime comparisons between the proposed method and existing approaches. Including this information would be valuable for assessing the method’s practicality in real-world applications.
- It would be beneficial to present qualitative results of the method on more challenging scenarios, such as the breakdancing sequences in the DAVIS dataset. Analyzing performance on these difficult cases would provide deeper insights into the task’s complexity and the robustness of the proposed approach.

---

> ### Author Rebuttal · Authors · 2025-07-31
>
> We thank the reviewer for their feedback, finding our work novel and clear.
>
> **Regarding runtime comparisons:**
> Over a total of 360 individual runs each taken place on WAN 2.1 based our implementation with floating point 16 precision. Each run has been executed in a single H100 gpu. the configuration of each run is as follows:
> * number of optimization steps = 5
> * height of video = 480
> * width of video = 832
> * number of frames = 49
> * number of inference steps = 50
>
> According to this configuration, you can find the runtime information in the table below:
>
> |                   |        Conmo        |                 SMM | DitFlow (Rope)      | DitFlow (Latent)    | Moft                | GWTF | Ours |
> | :---------------- | :-----------------: | ------------------: | ------------------- | ------------------- | ------------------- | ---- |  ---- |
> | Runtime (seconds) | 150.666 ± 3.317 | 107.281 ± 4.562 | 104.405 ± 2.056 | 105.126 ± 2.739 | 119.411 ± 3.847 | 101.342 ± 3.337 | 109.231 ± 3.112 |
>
> As shown in the table, all methods except ConMo have comparable runtimes, with our method running at 109.2 ± 3.1 seconds. While not the fastest, it remains within the typical range observed across recent approaches, balancing efficiency with improved realism. We appreciate the reviewer’s suggestion regarding runtime analysis. We will add the comparison table and a brief discussion in the revised manuscript.
>
> **Regarding more challenging datasets:** Our method is also successful in more challenging scenarios beyond the DAVIS dataset, such as RealCamVid [1]. However, due to this year’s new rules, we cannot provide links or a PDF to showcase our results. You will be able to see the additional experiments in the updated Supplementary, in case of acceptance, thank you for your understanding. To provide evidence that our method can handle camera motion and multiple object cases better compared to current baselines, we have randomly sampled 20 videos from RealCamVid [1] dataset ensuring they contain camera motion, multi object and relatively harder samples compared to DAVIS. We have generated 2 prompts per video using the same approach described in the main paper. The computed metrics for these models is as follows:
>
> | Method                  | MF ↑    | CLIP ↑   | FTD ↓ |
> |-------------------------|--------:|---------:|--------------:|
> | *Ours*                | 0.7019  | 0.2141   | 0.1697        |
> | *GWTF*                | 0.5796  | 0.2087   | 0.2072        |
> | *ConMo*               | 0.5973  | 0.2014   | 0.2800        |
> | *SMM*                 | 0.5717  | 0.2118   | 0.2827        |
> | *DiTFlow (RoPE)*      | 0.5686  | 0.2118   | 0.2991        |
> | *DiTFlow (latent)*    | 0.5693  | 0.2099   | 0.2962        |
> | *MOFT*                | 0.5778  | 0.2095   | 0.2873        |
>
> As it can be seen, our model surpasses all of its competitors on text and motion alignment in this dataset.
>
> In addition, regarding the **breakdance example in the DAVIS dataset**, our method performs significantly better than the others. We appreciate the reviewer’s suggestion and will add harder examples in the revised manuscript.
>
> [1] Zheng, Guangcong, et al. "Realcam-vid: High-resolution video dataset with dynamic scenes and metric-scale camera movements." arXiv preprint arXiv:2504.08212 (2025).

---

> > ### Comment · Reviewer_d2Er · 2025-08-04
> >
> > I appreciate the authors for their rebuttal. My concerns have been addressed, and I am looking forward to seeing the harder qualitative results examples in the revised manuscript. I would like to keep my positive score.

---

### Official Review · Reviewer_9b3n · 2025-06-28

**Clarity:** 3
**Significance:** 2
**Originality:** 3
**Rating:** 4
**Confidence:** 3

**Summary:**

The authors propose a training-free motion transfer methods. The core idea of the method lies in leveraging optical flow warp RoPE embeddings for the target video. To improve results, the authors further fine-tune the embeddings at test time using a flow matching objective regularized with a phase regularization term. A new evaluation metric is proposed and a good amount of qualitative results are provided that show good output quality. Quantitative evaluation places the method at the level of the current state of the art.

**Questions:**

- The method seems to improve performance qualitatively over the current state of the art. This is not captured by the contributed metric. It is thus unclear where the proposed method stands with respect to the state of the art, and whether the proposed metric is a good metric. This prevents the reader both from understanding quality of the proposed method and whether FTD is an informative metric, making the significance of the work unclear. I think both these points could be addressed with user studies showing that: 1. Overall the proposed method is preferred to the current state of the art despite having poorer motion alignment quality as shown by FTD, 2. FTD correlates to motion alignment quality better than MF. If these two points can be shown I would reconsider my score.

**Ethical Concerns:**

["NO or VERY MINOR ethics concerns only"]

**Final Justification:**

The rebuttal clarified my doubts regarding the proposed metrics and performance of the method with respect to baselines. The inclusion of the user studies for both evaluation metrics and proposed method quality strengthen the evaluation and demonstrate the significance of the work. I thus raise my score to recommend acceptance.

**Limitations:**

- Limitations are adequately discussed

**Paper Formatting Concerns:**

- Usage of bolding rather than italic in Tab. 1 would improve readability

**Quality:**

2

**Strengths And Weaknesses:**

QUALITY
- The paper shows a good qualitative evaluation section as part of the supplementary which shows good method performance with respect to the current state of the art
- Authors propose as a contribution Frechet Trajectory Distance (FTD) as a more reliable of motion alignment than MF. In Tab. 1, however, GWTF outperforms the proposed method in this very metric. This raises questions on whether either the proposed metric contribution is not representative of motion quality or whether the proposed method does not beat the baseline in this regard. Looking at the qualitative results, it seems the proposed method produces more realistic results, while GWTF more strictly adheres to the conditioning motion. I'd like the authors to clarify this point
- Given the uncertainty arisen by automated quantitative metrics, a user study would have enhanced thoroughness of the evaluation section, validating both the effectiveness of the proposed method and metric contributions


CLARITY
- The paper is well written and easy to follow. Algorithms are presented to clarify the implementation of the main method components. The supplementary material describes training hyper parameters and the proposed FTD metric implementation in detail.


SIGNIFICANCE
- Motion transfer is a an established research direction and the proposed method seemingly improves over the current state of the art, at least qualitatively. However, the incongruent performance of the proposed method with respect to the contributed FTD metric, compounded by the absence of a user study evaluating metric and method effectiveness, make the effectiveness of the method and proposed metric unclear to the reader.

ORIGINALITY
- The idea of transfering motion by altering RoPE embeddings is interesting and the authors explore it deeply, showing the necessity of embedding fine-tuning and phase regularization. This contribution is original to the best of my knowledge

---

> ### Author Rebuttal · Authors · 2025-07-31
>
> We thank the reviewer for their feedback, finding our work easy to follow, original, and interesting.
>
> **Regarding FTD:** We agree with the observation that RoPECraft produces more realistic results, whereas GWTF adheres more strictly to the conditioning motion. This likely explains why GWTF slightly outperforms RoPECraft on the FTD metric, as noted by the Reviewer. Since FTD is specifically designed to evaluate motion alignment rather than visual realism, the difference in scores reflects this trade-off.  However, our results also show that strict adherence to the conditioning motion does not always lead to better outcomes.  GWTF has higher adherence to the reference motion, we observe that this is not an advantage for GWTF, leading to distortions and conflicts with the text prompt. For example, when generating a flock of animals from a single-object reference video, GWTF often overfits to the single instance, producing unrealistic and distorted outputs (see “CogVideoX results” on the website, in the supplementary ZIP).
>
> We believe that both motion adherence (e.g., FTD) and realism/naturalness (e.g., user study and CD-FVD scores) are essential for a holistic evaluation of motion transfer methods. This is a very important point, and we will include this discussion in the final manuscript, thank you for the suggestion.
>
> **Regarding user study:** As suggested, we conducted a user study with 21 participants to evaluate both realism and proposed metrics alignment with human preferences. The user study is conducted as follows:
>
> For sample selection, we randomly sampled 20 prompts and reference videos from DAVIS and their custom prompts described in the paper, and generated outputs using all compared methods (including ours), and presented users with the resulting videos for each prompt-reference pair. Then, we calculated the FTD and MF score for each generated video-ground truth video pair. For reference, some portion of the data is given below:
>
> | Data | GWTF | RoPECraft | … |  |
> |------|------|-----------|---|--|
> | **Prompt**: A woman wearing a black dress walks down a worn wooden dock along the edge of a misty lake at dusk  **GT video**: lucia.mp4 | FTD and MF scores | FTD and MF scores | FTD and MF scores |  |
> | **Prompt**: A woman rides a bike down a wooden dock alongside a serene lake at sunrise.  **GT video**: bmx-trees.mp4 | FTD and MF scores | FTD and MF scores | MF scores |  |
>
> In the first part of the survey, users were shown the 7 generated videos per prompt and asked to select the top 3 that best matched the motion in the reference video. Participants were instructed to **focus solely on motion alignment, ignoring visual quality.** This section aimed to compare how well FTD and MF align with human perception of motion. For each prompt and video sample the procedure is as follows:
>
> * Provide all 7 video samples to users, for each question (10 in total).
> * Users pick top-3 samples among 7.
> * For each video, we also get the subset of ground-truth FTD and MF sorted scores.
> * We compare their user and ground-truth match, and assign percentages, to get a sub-table for each user.
> * We average and normalize each sub-table of each user to get the final table.
>
> For Part 1, on the average of all prompts and all users, the results are as follows:
> |              | Alignment with FTD (%) | Alignment with MF (%) |
> |--------------|------------------------|------------------------|
> | 1st choice   | **25**                 | 11                     |
> | 2nd choice   | **17**                 | 17                     |
> | 3rd choice   | **19**                 | 11                     |
>
> The table clearly reveals that our proposed FTD metric is more aligned with human perception, further suggesting its success.
>
> **In the second section of the survey,** users were asked to order the best 3 among the outputs based on visual quality, text prompt alignment, and motion alignment. This aimed to evaluate overall method quality. For each prompt and video sample the procedure is as follows:
>
> * Provide all 7 video samples to users, for each question (10 in total).
> * Users pick top-3 samples among 7.
> * Average all 3 choices in themselves (each row would sum up to 100%)
>
> For Part 2, on the average of all prompts and all users, the results are as follows:
>
> |               | **RoPECraft** | **GWTF** | **SMM** | **MOFT** | **ConMo** | **DitFlow (Latents)** | **DitFlow (RoPE)** |
> |---------------|---------------|----------|---------|----------|-----------|------------------------|---------------------|
> | **1st choice** | **30%**       | 10%      | 13%     | 19%      | 10%       | 12%                   | 6%                  |
> | **2nd choice** | **23%**       | 12%      | 14%     | 11%      | 13%       | 13%                   | 14%                 |
> | **3rd choice** | **13%**       | 11%      | 22%     | 13%      | 16%       | 13%                   | 12%                 |
>
>
> The table shows that our proposed method, RoPECraft, is consistently rated as more visually appealing, accounting for visual quality, text prompt alignment, and motion alignment all at the same time.  RoPECraft was consistently preferred over other methods, indicating that **our approach is better than the current literature in human perception.**
>
> The results reveal two key insights: (1) FTD correlates better than MF in evaluating motion fidelity, and (2) RoPECraft is consistently rated as more visually appealing while also achieving good motion alignment.
>
> We would also like to again point out why FTD may be preferred over MF. FTD can be more reliable than MF for evaluating motion alignment because it considers the entire motion trajectory over time, capturing the global structure and temporal consistency of motion. Unlike MF, which focuses on frame-to-frame displacements and overlooks long-term trajectory patterns, FTD evaluates the overall shape of the motion path, making it sensitive to deviations across multiple frames. FTD also effectively handles occlusions by reassigning or discarding invalid trajectories, ensuring that only valid motion paths contribute to the score. In contrast, MF does not account for occlusions or tracking errors, leading to potentially misleading results. Furthermore, while MF normalizes frame-wise displacements and ignores their magnitude, FTD preserves such variations, making it more robust in capturing complex motion dynamics. Overall, we argue that FTD offers a more holistic and accurate reflection of motion similarity, especially in dynamic or occlusion-heavy scenes.

---

> > ### Comment · Reviewer_9b3n · 2025-08-03
> >
> > I thank the authors for their rebuttal.
> > The rebuttal clarified my doubts regarding the proposed metrics and performance of the method with respect to baselines. The inclusion of the user studies for both evaluation metrics and proposed method quality strengthen the evaluation and demonstrate the significance of the work. I thus raise my score to recommend acceptance.

---

> > > ### Author Response · Authors · 2025-08-04
> > >
> > > We would like to thank the reviewer for taking the time to evaluate both our submission and rebuttal. We are pleased to hear that your concerns regarding our proposed metric have been resolved and sincerely appreciate your positive assessment.

---

### Official Review · Reviewer_Sq4g · 2025-07-01

**Clarity:** 3
**Significance:** 3
**Originality:** 3
**Rating:** 4
**Confidence:** 3

**Summary:**

RoPECraft is a training-free video motion transfer technique for diffusion transformers that works by directly manipulating their rotary positional embeddings (RoPE). It warps RoPE’s complex-exponential tensors using dense optical flow from a reference video to encode motion into generation, then refines these embeddings during denoising via a flow-matching loss that aligns predicted and target velocities. Benchmark evaluations show RoPECraft surpasses recent state-of-the-art methods both qualitatively and quantitatively.

**Questions:**

N.A.

**Ethical Concerns:**

["NO or VERY MINOR ethics concerns only"]

**Final Justification:**

The authors have addressed my questions and concerns, I will keep my positive score.

**Limitations:**

Yes.

**Paper Formatting Concerns:**

N.A.

**Quality:**

3

**Strengths And Weaknesses:**

Strengths:
1. The paper is well written and easy to follow.

2. The idea of modifying rotary positional embeddings (RoPE) for motion transfer is insightful and interesting.

Questions:

1. All showcased examples and supplementary results focus on single-object motion transfer. Can RoPECraft handle multiple-object motion transfer?

2. During motion transfer, the background often changes significantly, even though the prompt did not include background alteration. Can the method preserve the original background while transferring only the object’s motion?

3. The column indices referenced in Figure 2’s caption appear misleading—please correct them in the final version.

---

> ### Author Rebuttal · Authors · 2025-07-31
>
> We thank the reviewer for their positive feedback and for recognizing our work as insightful, interesting, and well presented.
>
> **Regarding multi-object cases:** RoPECraft is capable of handling multi-object motion transfer. However, due to this year’s new rules, we cannot provide links or a PDF to showcase our results. You will be able to see the additional experiments in the updated Supplementary, in case of acceptance, thank you for your understanding. To provide evidence that our method can handle camera motion and multiple object cases better compared to current baselines, we have randomly sampled 20 videos from RealCamVid [1] dataset ensuring they contain camera motion, multi object and relatively harder samples compared to DAVIS. We have generated 2 prompts per video using the same approach described in the main paper. The computed metrics for these models is as follows:
>
> | Method                  | MF ↑    | CLIP ↑   | FTD ↓ |
> |-------------------------|--------:|---------:|--------------:|
> | ***Ours***                | **0.7019**  | **0.2141**   | **0.1697**        |
> | *GWTF*                | 0.5796  | 0.2087   | 0.2072        |
> | *ConMo*               | 0.5973  | 0.2014   | 0.2800        |
> | *SMM*                 | 0.5717  | 0.2118   | 0.2827        |
> | *DiTFlow (RoPE)*      | 0.5686  | 0.2118   | 0.2991        |
> | *DiTFlow (latent)*    | 0.5693  | 0.2099   | 0.2962        |
> | *MOFT*                | 0.5778  | 0.2095   | 0.2873        |
>
> As it can be seen, our model surpasses all of its competitors on text and motion alignment in this dataset.
>
> **Regarding preserving background:** The goal of motion transfer is to transfer the motion from a reference video to a newly generated one, guided by a text prompt. While the prompt may not always specify the background, DiT models (owing to their attention mechanisms and training data) naturally generate coherent and realistic backgrounds.
>
> That said, by applying flow-matching optimization for longer and using background masks, it is possible to preserve the original background while transferring only the motion. In fact, we verified that it works with our parameters in Supplementary A.3; but with the mask and by changing s=30 and t=20 for this sub-task. We will include the additional results to the Supplementary, thank you for suggesting this.
>
> **Regarding typos:** In Fig. 2, the caption for Column 4 should read Column 5. We will go through the manuscript and fix the errors, thank you.
>
> [1] Zheng, Guangcong, et al. "Realcam-vid: High-resolution video dataset with dynamic scenes and metric-scale camera movements." arXiv preprint arXiv:2504.08212 (2025).

---

### Official Review · Reviewer_PryD · 2025-07-03

**Clarity:** 3
**Significance:** 3
**Originality:** 3
**Rating:** 4
**Confidence:** 3

**Summary:**

This paper studies the task of performing video-level motion transfer by using priors from video DiT without further test-time training or fine-tuning. It is based on the idea of a prior work called DiTFlow, yet it eliminates the need for optimizing the latent space. The key idea is to infer motion information from estimated optical flows and apply those hints to RoPE positional encodings. A metric called FTD is proposed to evaluate the proposed method.

**Questions:**

- In almost all comparisons, MOFT / DiTFlow / ConMo have an opposite direction of motion. Is it because of the limitations of the baselines, or are there any issues in re-implementations?
- Is it possible to provide a theoretical analysis of the proposed methods?
- A user study might be useful to see both the performance of the proposed pipeline and how the proposed metric aligns with human preferences.
- How does the proposed method approach videos with camera motion?

**Ethical Concerns:**

["NO or VERY MINOR ethics concerns only"]

**Final Justification:**

The rebuttal has addressed most of my initial concerns. I will keep a positive score.

**Limitations:**

This paper relies on a pre-trained optical flow estimator. If the optical flow extracted is not accurate enough, the proposed method will likely fail.

**Quality:**

3

**Strengths And Weaknesses:**

**Strengths**

- The studied problem is interesting and important. Achieving motion transfer between different subjects has a lot of applications in various communities.

- The proposed motion transfer method is training-free, which eliminates the need for additional fine-tuning or test-time training.
- The qualitative results show non-trivial improvements compared to previous methods.
- The ablation studies are informative to show the effectiveness of different components.

**Weaknesses**

- There is no theoretical guarantee on the correctness of the motion-modulated RoPE algorithm. All results are empirical.
- The proposed approach only applies to video generative models that use RoPE as their positional encoding in DiT.
- There is no evidence on how good the proposed FTD metric is. Is it possible to perform a user study to demonstrate how users' preferences align with the proposed FTD metric?

---

> ### Author Rebuttal · Authors · 2025-07-31
>
> We thank the reviewer for their positive feedback and for highlighting the significance of our problem setting, the advantages of our training-free approach, the qualitative improvements over prior work, and the value of our ablation studies.
>
> **Regarding the motion directions of other methods:** Our experiments with MOFT, DiTFlow, and ConMo are based directly on the official DiTFlow [CVPR25] codebase. We did not re-implement these methods but used the provided code implementation in Wan 2.1 as-is. We have carefully double-checked the results and can confirm that the observed motion direction differences stem from limitations of the baseline methods, not from any implementation issues.
>
> While the motion direction is generally correct across methods, we observe that object orientation is often incorrect in other competing methods. This can be more clearly seen with the background motion, which suggests that motion is being transferred to incorrect subjects. Our flow-matching and phase constraints address these issues, leading to more accurate motion transfer. We will add this discussion to the final manuscript, thank you for pointing it out.
>
> **Regarding theoretical analysis and applicability to RoPE-based models:** We agree that our study is primarily empirical and specifically applies to models using RoPE as their positional encoding within DiT architectures. However, we believe this does not diminish the contribution of our work. RoPE is increasingly adopted in modern generative models, and our experimental insights provide a timely and valuable foundation for future research. There is growing interest in exploring RoPE-based mechanisms in recent literature. For example, recent work has extended its use to camera matrices [1] or there have been recent works that aim to update RoPE for longer video generation [2] or their integration to the video models [3]. Similarly, our approach (modifying RoPE tensors directly with flow values) may inspire further advancements, such as consistent long video generation or video editing.
>
>  **Regarding user study:** As suggested, we conducted a user study with 21 participants to evaluate both realism and proposed metrics alignment with human preferences. The user study is conducted as follows:
>
> For sample selection, we randomly sampled 20 prompts and reference videos from DAVIS and their custom prompts described in the paper, and generated outputs using all compared methods (including ours), and presented users with the resulting videos for each prompt-reference pair. Then, we calculated the FTD and MF score for each generated video-ground truth video pair. For reference, some portion of the data is given below:
>
> | Data | GWTF | RoPECraft | … |  |
> |------|------|-----------|---|--|
> | **Prompt**: A woman wearing a black dress walks down a worn wooden dock along the edge of a misty lake at dusk  **GT video**: lucia.mp4 | FTD and MF scores | FTD and MF scores | FTD and MF scores |  |
> | **Prompt**: A woman rides a bike down a wooden dock alongside a serene lake at sunrise.  **GT video**: bmx-trees.mp4 | FTD and MF scores | FTD and MF scores | MF scores |  |
>
> In the first part of the survey, users were shown the 7 generated videos per prompt and asked to select the top 3 that best matched the motion in the reference video. Participants were instructed to **focus solely on motion alignment, ignoring visual quality.** This section aimed to compare how well FTD and MF align with human perception of motion. For each prompt and video sample the procedure is as follows:
>
> * Provide all 7 video samples to users, for each question (10 in total).
> * Users pick top-3 samples among 7.
> * For each video, we also get the subset of ground-truth FTD and MF sorted scores.
> * We compare their user and ground-truth match, and assign percentages, to get a sub-table for each user.
> * We average and normalize each sub-table of each user to get the final table.
>
> For Part 1, on the average of all prompts and all users, the results are as follows:
> |              | Alignment with FTD (%) | Alignment with MF (%) |
> |--------------|------------------------|------------------------|
> | 1st choice   | **25**                 | 11                     |
> | 2nd choice   | **17**                 | 17                     |
> | 3rd choice   | **19**                 | 11                     |
>
> The table clearly reveals that our proposed FTD metric is more aligned with human perception, further suggesting its success.
>
> **In the second section of the survey,** users were asked to order the best 3 among the outputs based on **visual quality, text prompt alignment, and motion alignment**. This aimed to evaluate overall method quality. For each prompt and video sample the procedure is as follows:
>
> * Provide all 7 video samples to users, for each question (10 in total).
> * Users pick top-3 samples among 7.
> * Average all 3 choices in themselves (each row would sum up to 100%)
>
> For Part 2, on the average of all prompts and all users, the results are as follows:
>
> |               | **RoPECraft** | **GWTF** | **SMM** | **MOFT** | **ConMo** | **DitFlow (Latents)** | **DitFlow (RoPE)** |
> |---------------|---------------|----------|---------|----------|-----------|------------------------|---------------------|
> | **1st choice** | **30%**       | 10%      | 13%     | 19%      | 10%       | 12%                   | 6%                  |
> | **2nd choice** | **23%**       | 12%      | 14%     | 11%      | 13%       | 13%                   | 14%                 |
> | **3rd choice** | **13%**       | 11%      | 22%     | 13%      | 16%       | 13%                   | 12%                 |
>
>
> The table shows that our proposed method, RoPECraft, is consistently rated as more visually appealing, accounting for visual quality, text prompt alignment, and motion alignment all at the same time.  RoPECraft was consistently preferred over other methods, indicating that **our approach is better than the current literature in human perception.**
>
> The results reveal two key insights: (1) FTD correlates better than MF in evaluating motion fidelity, and (2) RoPECraft is consistently rated as more visually appealing while also achieving good motion alignment.
>
> We would also like to again point out why FTD may be preferred over MF. FTD can be more reliable than MF for evaluating motion alignment because it considers the entire motion trajectory over time, capturing the global structure and temporal consistency of motion. Unlike MF, which focuses on frame-to-frame displacements and overlooks long-term trajectory patterns, FTD evaluates the overall shape of the motion path, making it sensitive to deviations across multiple frames. FTD also effectively handles occlusions by reassigning or discarding invalid trajectories, ensuring that only valid motion paths contribute to the score. In contrast, MF does not account for occlusions or tracking errors, leading to potentially misleading results. Furthermore, while MF normalizes frame-wise displacements and ignores their magnitude, FTD preserves such variations, making it more robust in capturing complex motion dynamics. Overall, we argue that FTD offers a more holistic and accurate reflection of motion similarity, especially in dynamic or occlusion-heavy scenes.
>
>
> **Regarding camera motion:** Our method effectively transfers camera motion. However, due to this year’s new rules, we cannot provide links or a PDF to showcase our results. You will be able to see the additional experiments in the updated Supplementary, in case of acceptance, thank you for your understanding. To provide evidence that our method can handle camera motion and multiple object cases better compared to current baselines, we have randomly sampled 20 videos from RealCamVid [4] dataset ensuring they contain camera motion, multi object and relatively harder samples compared to DAVIS. We have generated 2 prompts per video using the same approach described in the main paper. The computed metrics for these models is as follows:
>
> | Method                  | MF ↑    | CLIP ↑   | FTD ↓ |
> |-------------------------|--------:|---------:|--------------:|
> | ***Ours***                | **0.7019**  | **0.2141**   | **0.1697**        |
> | *GWTF*                | 0.5796  | 0.2087   | 0.2072        |
> | *ConMo*               | 0.5973  | 0.2014   | 0.2800        |
> | *SMM*                 | 0.5717  | 0.2118   | 0.2827        |
> | *DiTFlow (RoPE)*      | 0.5686  | 0.2118   | 0.2991        |
> | *DiTFlow (latent)*    | 0.5693  | 0.2099   | 0.2962        |
> | *MOFT*                | 0.5778  | 0.2095   | 0.2873        |
>
> As it can be seen, our model surpasses all of its competitors on text and motion alignment in this dataset.
>
> [1] Li et al., Cameras as Relative Positional Encoding, 2025.
>
> [2] RIFLEx: A Free Lunch for Length Extrapolation in Video Diffusion Transformers, ICML 25
>
> [3] VideoRoPE: What Makes for Good Video Rotary Position Embedding?, ICML 25, Oral
>
> [4] Zheng, Guangcong, et al. "Realcam-vid: High-resolution video dataset with dynamic scenes and metric-scale camera movements." arXiv preprint arXiv:2504.08212 (2025).

---

> > ### Comment · Reviewer_PryD · 2025-08-05
> >
> > Thanks for your rebuttal! I'm keeping my initial positive score.

---

### Note · Authors · 2025-08-14

We would like to thank the reviewers and the AC for their time and effort. We appreciate the constructive feedback we received, which helped us strengthen the paper. We are pleased that our rebuttal addressed the reviewers’ concerns, and we are encouraged by their positive assessments.

---

### Decision · Program_Chairs · 2025-09-17

**Decision:**

Accept (poster)

**Comment:**

This paper was reviewed by four experts in the field. Most concerns are solved in the rebuttal and discussions. The paper finally received borderline positive reviews with 4 Borderline Accepts.

This paper proposes a training-free video motion transfer technique by directly manipulating their RoPE.
It first warps RoPE’s complex-exponential tensors using optical flow from the input video to provide motion guidance.
It then optimizes the embeddings at test time via a flow-matching loss that aligns predicted and target input video's velocitie, and a phase
constraint regularization to ensure temporal coherence. A metric called FTD is proposed to evaluate the proposed method.
Experimental results show that the proposed method outperforms other baselines.

**strengths**
- the task has a lot of applications in various communities
- a training-free method, the idea of warp  rotary positional embeddings for motion transfer is simple yet effective
- non-trivial improvements compared to previous methods
- well written and easy to follow

**weaknesses**
- there is no evidence on how the proposed metric FTD aligns with the human preference, whether FTD is an informative metric
- insufficient experimental validation: missing performance on camera motion transfer, missing results on multiple-object motion transfer, missing runtime comparisons


According to the reviews, the reviewers acknowledge the paper's good results and novel idea. The missing experimental validations are provided in the rebuttal. The AC appreciates the very comprehensive user study to validate the effectiveness of FTD and the superiority of the proposd method over other baselines. Following the rebuttal, reviewers have provided relatively positive ratings for the paper. In the revision, the authors are suggested to incorporate the reviewers' feedback, to include the required experimental results.